# Calcium Phosphate Bions Cause Intimal Hyperplasia in Intact Aortas of Normolipidemic Rats through Endothelial Injury

**DOI:** 10.3390/ijms20225728

**Published:** 2019-11-15

**Authors:** Daria Shishkova, Elena Velikanova, Maxim Sinitsky, Anna Tsepokina, Olga Gruzdeva, Leo Bogdanov, Anton Kutikhin

**Affiliations:** Division of Experimental and Clinical Cardiology, Research Institute for Complex Issues of Cardiovascular Diseases, 6 Sosnovy Boulevard, Kemerovo 650002, Russia; shishkovadk@gmail.com (D.S.); veliea@kemcardio.ru (E.V.); max-sinitsky@rambler.ru (M.S.); cepoav@kemcardio.ru (A.T.); o_gruzdeva@mail.ru (O.G.); bogdanovleone@gmail.com (L.B.)

**Keywords:** bions, calciprotein particles, calcium phosphate, hydroxyapatite, cardiovascular disease, blood flow, intimal hyperplasia, neointima, endothelial dysfunction, endothelial injury

## Abstract

Calcium phosphate bions (CPBs) are formed under blood supersaturation with calcium and phosphate owing to the mineral chaperone fetuin-A and representing mineralo-organic particles consisting of bioapatite and multiple serum proteins. While protecting the arteries from a rapid medial calcification, CPBs cause endothelial injury and aggravate intimal hyperplasia in balloon-injured rat aortas. Here, we asked whether CPBs induce intimal hyperplasia in intact rat arteries in the absence of cardiovascular risk factors. Normolipidemic Wistar rats were subjected to regular (once/thrice per week over 5 weeks) tail vein injections of either spherical (CPB-S) or needle-shaped CPBs (CPB-N), magnesium phosphate bions (MPBs), or physiological saline (*n* = 5 per group). Neointima was revealed in 3/10 and 4/10 rats which received CPB-S or CPB-N, respectively, regardless of the injection regimen or blood flow pattern in the aortic segments. In contrast, none of the rats treated with MPBs or physiological saline had intimal hyperplasia. The animals also did not display signs of liver or spleen injury as well as extraskeletal calcium deposits. Serum alanine/aspartate transaminases, interleukin-1β, MCP-1/CCL2, C-reactive protein, and ceruloplasmin levels did not differ among the groups. Hence, CPBs may provoke intimal hyperplasia via direct endothelial injury regardless of their shape or type of blood flow.

## 1. Introduction

Calcium phosphate bions (CPBs), alternatively termed calciprotein particles, are mineralo-protein nano-sized complexes generated in the blood as a result of its supersaturation with calcium and/or phosphate [1,2]. Molecular insights into the inhibition of extraskeletal calcification suggest that the mineral chaperones fetuin-A, albumin, and other acidic serum proteins are responsible for the controlled aggregation of excessive calcium and phosphate into the CPBs [3,4]. In particular, fetuin-A is attributable to the shielding of nascent mineral nuclei, thereby hampering growth and precipitation of calcium phosphate crystals [4,5]. Subsequent studies confirmed that CPBs represent an elegant mechanism governing the mineral homeostasis and preventing ectopic mineralization [1,6].

While protecting the body from medial arterial calcification, a formidable condition causing rapid and irreversible mechanical incompetence of the blood vessels [7,8], CPBs nevertheless act as a double-edged sword. Despite clinical investigations in this field having been primarily focused on kidney transplant recipients [9,10,11] and patients with end-stage renal disease [12] or pre-dialysis chronic kidney disease (CKD) [13], an elevated serum propensity to form CPBs was also observed in non-CKD patients with arterial hypertension, a major risk factor of coronary artery disease, in comparison with healthy blood donors [14]. Further, an increased CPB count in the serum is associated with the progression of stable angina to acute coronary syndrome and also with the lipid and total plaque volume in patients without detectable CKD [15]. Our unpublished data also indicated that serum levels of CPBs are higher in patients with coronary artery disease requiring bypass surgery than in healthy individuals.

Incubation of endothelial cell cultures with CPBs provoked lysosome-dependent cell death accompanied by the augmented release of pro-inflammatory cytokines interleukin-6 and interleukin-8; these effects were more pronounced if needle-shaped CPBs (CPB-N) were applied instead of spherical CPBs (CPB-S) [6]. Moreover, intravenous administration of CPBs entailed the development of intimal hyperplasia and adventitial inflammation upon balloon injury in Wistar rats [6]. However, in this model, CPBs enhanced pre-existing vascular injury rather than causing it per se. To conclusively demonstrate the clinical relevance of CPBs in an experimental setting, these particles need to be regularly injected into the circulation without any concurrent cardiovascular risk factors. The latter model would therefore be able to simulate the scenario of a healthy patient having an increased serum propensity to generate CPBs as the only factor detrimental to the vascular homeostasis.

Besides pathological states, such as dyslipidemia, hyperglycemia, metabolic syndrome or CKD, atherosclerosis is also influenced by a complex arterial geometry being characteristic of the segments with turbulent or oscillatory blood flow (e.g., aortic arch, bifurcations, and branching points) [16,17,18]. Triggers of endothelial injury such as oxidized low-density lipoproteins [19,20] or high serum glucose [21,22] frequently promote liver fibrosis [19,20,21,22] and contribute to the systemic inflammatory response [23,24,25] which are among the risk factors of atherosclerosis aggravating deleterious consequences of endothelial injury [26,27,28].

Here, we investigated whether a consistent intravenous administration of CPBs to normolipidemic rats induced the development of intimal hyperplasia in intact aortas and further examined whether this process was affected by an aortic anatomy as well as the shape and systemic side effects of CPBs. We found that regular infusions of CPBs resulted in neointimal formation in 30–40% of the rats regardless of particle shape, blood flow pattern, or injection regimen, probably through a direct endothelial injury.

## 2. Results

To study whether the pathogenic effects of CPBs are sufficiently significant to cause endothelial injury without any background factors determining cardiovascular risk, we performed regular intravenous injections (1 or 3 times per week during the 5 weeks) of either CPB-S, CPB-N, innocuous magnesium phosphate bions (MPB) [29], and physiological saline, used as a vehicle control, into the tail vein of normolipidemic Wistar rats. At the end of the 5th week, the animals were sacrificed with the subsequent excision of the segments with a turbulent or oscillatory flow (aortic arch and bifurcation) and laminar flow (descending aorta) followed by histological analysis and immunofluorescence examination.

We earlier documented an 80%/90% prevalence of intimal hyperplasia in the abdominal aortas of Wistar rats upon the combination of a balloon angioplasty and intravenous administration of CPB-S, or CPB-N but not MPB [29]. Strikingly, we found signs of intimal hyperplasia defined as abundant extracellular matrix (ECM) and multiple randomly oriented polygonal cells (Figure 1A) in 30% (3/10) and 40% (4/10) of rats subjected to CPB-S and CPB-N, respectively (Figure 1B). Neointima was clearly detectable both in hematoxylin and eosin and Russell–Movat’s pentachrome staining (Figure 1A). In agreement with our previous findings [6], aortas of rats treated with CPB were devoid of calcium deposits, confirming a protective role of these particles against arterial calcification (Figure 1A). No association of the intimal hyperplasia with the turbulent/oscillatory or laminar flow was identified, suggesting that CPBs evince their effects regardless of the blood flow pattern; similar results were obtained with respect to the injection regimen (Figure 1B,C). None of the rats which received MPB or physiological saline had alterations of a vascular architecture.

With the aim of comparing the features of CPB-induced neointima in this animal model to other scenarios, we conducted an immunophenotyping utilizing CD31/CD34, CD31/α-smooth muscle actin (αSMA), αSMA/collagen IV, and αSMA/vimentin stainings to assess re-endothelialization, neointimal cell populations, production of ECM, and phenotypic switch of smooth muscle cells, respectively. Both neointima and intact aortas were lined with a monolayer of CD31^+^CD34^−^αSMA^−^ cells, indicative of mature endothelial phenotype without ongoing re-endothelialization, neovascularization, or endothelial-to-mesenchymal transition (Figure 2A). In concert with the findings from apolipoprotein E (ApoE)^−/−^ or low-density lipoprotein receptor (LDLR)^−/−^ mice inherently prone to the development of atherosclerosis and human atherosclerotic plaques [30], CPB-induced neointima consisted of CD31^−^αSMA^+^ cells producing ECM protein collagen IV (Figure 2B,C). However, the mentioned cell populations did not express vimentin, a marker of a synthetic vascular smooth muscle cell phenotype [31,32] (Figure 2D). Taken together, these results testified to the formation of neointima in intact aortas of normolipidemic rats upon the intravenous administration of CPB.

We then sought to confirm that the development of intimal hyperplasia in rat aortas is attributable to the direct endothelial injury but not side effects related to hepatotoxicity, splenotoxicity, or systemic inflammation, all recognized as a confounding risk factors for the progression of atherosclerosis [26,27,28]. No signs of hepatocellular damage, liver fibrosis, or splenic injury was revealed upon hematoxylin and eosin (Figure 3A) and van Gieson staining (Figure 3B). Similar to the aortas, no calcium deposits were found in the liver or spleen of the experimental animals (Figure 3C).

Measurement of serum aspartate transaminase (AST) and alanine transaminase (ALT) which are well-established markers of liver injury, also did not reveal any statistically significant differences among the animal groups, corroborating the results of the histological examination (Figure 4).

Likewise, analysis of the serum levels of pro-inflammatory cytokines (interleukin (IL)-1β and monocyte chemoattractant protein 1/C-C motif chemokine ligand 2 (MCP-1/CCL2)) and acute phase proteins (C-reactive protein and ceruloplasmin) demonstrated no statistically significant intergroup differences (Figure 5) attesting to endothelial injury as the sole culprit of CPB-induced intimal hyperplasia.

We further asked which molecular mechanisms were responsible for the CPB-related endothelial injury. To address this issue, we conducted a dot blot profiling for 35 apoptosis-related proteins in primary human coronary artery endothelial cells (HCAECs) and human internal thoracic artery endothelial cells (HITAECs) treated with CPB-S over 4 h as well as control cells. In keeping with our previous Western blotting results [6], we found that cleaved caspase-3 and X-linked inhibitor of apoptosis protein (XIAP) notably increased in both HCAECs and HITAECs upon exposure to CPB-S, suggesting the involvement of an intrinsic apoptosis molecular pathway, while the level of inactive procaspase-3 was unaffected by the experimental conditions (Figure 6). The levels of other proteins were unchanged or inconsistently elevated/reduced among the cell lines (Figure 6).

In an attempt to denote specific signaling pathways mediating the deleterious effects of CPBs on endothelial cells, we profiled genes engaged in processes modulating endothelial dysfunction (Figure 7A). These included genes of cell adhesion molecules, cytokines and their receptors, scavenger receptors, oxidative stress enzymes, and endothelial-to-mesenchymal transition markers. Top differentially expressed genes (fold change >2 in both HCAECs and HITAECs exposed to CPB-S and CPB-N) were those encoding cytokines (*IL1B*, *IL6*, *IL8*, *IL23*), cell adhesion molecules (*ICAM1* and *SELP*), matrix metalloproteinase-2 (*MMP2*), and scavenger receptor class B member 3 (*CD36*). An enzyme-linked immunosorbent assay for IL-6 confirmed its augmented release by endothelial cells treated with CPBs (Figure 7B). The levels of IL-1b and IL-23 in cell culture supernatant were negligible while the concentrations of IL-8 did not differ significantly among the groups (data not shown). Hence, we propose that endothelial cell death triggered by CPBs is complemented by the concurrent excessive release of pro-inflammatory molecules such as IL-6.

## 3. Discussion

In the present study, we addressed the capability of CPBs to induce intimal hyperplasia in normolipidemic conditions, the absence of additional cardiovascular risk factors, and whether it was affected by CPBs’ shape and blood flow pattern in aortic segments. Albeit intravenous administration of CPBs led to the formation of neointima in 35% (7/20) of rats, it did not depend on the spherical or needle-shaped appearance of CPBs nor on the laminar/turbulent flow in corresponding vascular territories. Further experiments showed that the histological and immunohistochemical features of CPB-induced neointima were remarkably reminiscent of those that were observed in hyperlipidemic mice and plaques of patients with carotid or coronary atherosclerosis [30]. Measurement of the pro-inflammatory molecules in the serum and assessment of liver and spleen homeostasis did not reveal any considerable alterations, thereby pointing at vascular injury as the ultimate cause of intimal hyperplasia upon regular intravenous injections of CPBs.

Fluorescence labeling of fetuin-A, a mineral chaperone mainly accountable for the generation of CPBs, during the artificial synthesis determined the serum half-life of CPBs as 45 min [33,34]. After being introduced into the circulation of the mice, CPBs were rapidly (within 2 min) internalized and, subsequently, eliminated (within 25–30 min) by resident liver macrophages (Kupffer cells) [33,34]. Besides Kupffer cells, CPB-Ns were visualized in liver sinusoidal endothelial cells 25 min post-injection [34]. The CPBs were also found in marginal zone macrophages of the spleen, although in a 1.5–2 fold lower amount than in the liver [33]. Here, we did not detect CPBs in either the liver or spleen of Wistar rats, possibly owing to the rapid clearance of these particles, yet the absence of calcium deposits and fibrosis by means of negative alizarin red S and van Gieson staining testified to the harmlessness of CPBs to these organs. The levels of serum transaminases were also within the reference range, confirming these findings. Therefore, we suggest that CPBs do not possess hepato/splenotoxicity, and their detrimental effects to the endothelium are not related to hepatocellular damage.

Stimulation of cytokine release by vascular smooth muscle cells [35,36,37] and macrophages [34,38,39] in the case of their incubation with CPBs in vitro has been well documented; nevertheless, similar experiments have not been performed on cultures of neutrophils or lymphocytes nor has serum from the animals treated with CPBs been tested for levels of pro-inflammatory molecules. Here, we evaluated the concentrations of potent cytokines (IL-1β and MCP-1/CCL2) and acute phase proteins (C-reactive protein and ceruloplasmin) which are broadly established as mediators of inflammation in rat serum [40,41,42]. As we did not reveal any alterations in the levels of these molecules among the rats subjected to CPB, MPB, or physiological saline tail vein injections over 5 weeks, we propose that circulation of CPBs in the blood vessels does not induce a systemic inflammatory response, at least in this model. Studies on other models possibly including hyperlipidemic ApoE^−/−^ or LDLR^−/−^ are required to prove this hypothesis.

Investigations of the molecular mechanisms responsible for the exact mechanism of triggering intimal hyperplasia provoked by CPB treatment found that these particles cause regulated death of primary human endothelial cells involving intrinsic apoptosis cascade (i.e., cleavage of caspase-9 and caspase-3 and reciprocal upregulation of XIAP). Incubation of the mentioned cell cultures with CPB promoted the release of IL-6, a pro-inflammatory cytokine engaged in the development of atherosclerosis [43,44]. These results, taken together with the data obtained by other groups from vascular smooth muscle cells [35,36,37] and macrophages [34,38,39], suggest that CPBs evoke local rather than systemic inflammation, thereby altering the paracrine signaling among different populations of vascular cells and contributing to the development of a pathological microenvironment. The mechanism behind the neointima formation seems to be the combination of crude endothelial injury and stimulation of pro-inflammatory signaling in situ.

Electron microscopy analysis revealed spherical hydroxyapatite nanoparticles in both calcific and healthy segments of coronary arteries and aorta [45] as well as in aorta and iliac arteries of the patients with end-stage renal disease (ESRD) [46,47], a major risk factor of cardiovascular events and death [48,49]. Serum of the patients suffering from ESRD, coronary artery disease, or arterial hypertension was found prone to CPB formation as compared with healthy blood donors [14]. Further, increased serum propensity to generate CPBs was associated with an adverse outcome in patients with pre-dialysis chronic kidney disease (CKD) [13] and ESRD [12] including kidney transplant recipients [9,10,11]. Patients with ESRD had elevated risk of all-cause and cardiovascular death, myocardial infarction, and peripheral artery disease in the case of enhanced formation of CPBs [9,10,11,12]. Augmented serum propensity to produce CPBs correlated with severe coronary artery calcification and its progression in patients with CKD stages 2 to 4 [50]; this was partially verified by a recent study showing that higher CPB serum levels are more frequently observed in patients with acute coronary syndrome as compared to those with stable angina (without pre-dialysis CKD or ESRD) and is associated with the total and lipid plaque volume [15]. Hence, serum CPB concentration might be considered as a surrogate marker of coronary atherosclerosis and coronary artery calcification.

An original flow cytometry technique employing combined staining by a bisphosphonate conjugated with an infrared fluorescent dye (OsteoSense 680EX) and a green fluorescent membrane intercalating dye PKH67 enabled direct detection of CPBs (defined as OsteoSense^+^PKH67^-^ events) in peritoneal dialysis effluent of the patients with ESRD [51] and serum of the patients with pre-dialysis CKD [52] or ESRD as well as healthy volunteers [53]. Another option is microplate-based dynamic light scattering which is both high-throughput and precise in measuring the hydrodynamic radius of the nanoparticles and has also been applied to detect CPBs in the serum of subjects with pre-dialysis CKD and healthy blood donors [54]. Electron and atomic force microscopy represent an alternative but low-throughput method for CPB visualization upon the centrifugation of the serum collected from the patients with ESRD [39,54,55,56].

Identification of CPBs as a potential biomarker or even a trigger of endothelial dysfunction and injury demands an approach for their elimination from the circulation after the clearance of excessive calcium and phosphate ions. In vitro, CPBs can be routinely decalcified using ethylenediaminetetraacetic acid (EDTA) disodium salt; the clinical efficiency of this approach in a cardiovascular setting was evaluated in the Trial to Assess Chelation Therapy (TACT, NCT00044213). The TACT included 1708 patients ≥50 years of age who received (*n* = 839) or did not (*n* = 869) 40 infusions containing 3 g disodium EDTA (1.5 g/L) over 30 weeks. Such a disodium EDTA administration regimen was associated with a 1.22, 1.69, and 1.92 fold lower risk of a primary composite endpoint (death from any cause, repeated myocardial infarction, stroke, coronary revascularization, or hospitalization for angina pectoris) in a general cohort [57], subgroup of patients with diabetes mellitus [58], and those having diabetes mellitus and peripheral artery disease, respectively [59]. Intriguingly, the peak CPB concentration in the blood of patients with diabetes mellitus was observed at postprandial 2 h (after breakfast and dinner) [60], suggesting a link between postprandial glycemia and CPB formation that may explain a pronounced decrease in the risk of an adverse cardiovascular outcome in this patient category. Another advantage of the abovementioned therapeutic regimen is its relative safety [61]. However, insufficient bioavailability (≈5%) of disodium EDTA taken orally [62] considerably limits its clinical use. Upcoming clinical trials TACT2 (NCT02733185) and TACT3a (NCT03982693) are aimed at investigating the efficiency of the indicated chelation therapy regimen specifically in diabetic patients with a prior myocardial infarction and in individuals suffering from diabetes and critical limb ischemia as a manifestation of severe peripheral atherosclerosis.

Here we have, for the first time, shown that CPBs are able to cause typical intimal hyperplasia per se through an injury of initially intact endothelium. Importantly, our data reinforce the promising results from the clinical investigations reporting an association of increased serum propensity for CPB formation or higher CPB count with elevated risk and advanced stages of cardiovascular disease. Taken together, the indicated findings underscore the putative importance of CPBs, particles arising in human blood due to the neutralization of excessive mineral ions by fetuin-A, albumin, and other acidic serum proteins for the development of endothelial dysfunction and initiation of atherosclerosis. Further transcriptomic and proteomic profiling of the endothelial cells cultured under the pulsatile flow conditions and treated with CPBs in vitro or isolated from the blood vessels exposed to the circulating CPBs in vivo may provide deeper insight into these processes to uncover the molecular basis of CPB-induced endothelial dysfunction.

## 4. Materials and Methods

### 4.1. Artificial Synthesis of Calcium Phosphate and Magnesium Phosphate Bions (MPBs)

To synthesize CPB-S or CPB-N, stock solutions of CaCl_2_ (21115, Sigma–Aldrich, St. Louis, MO, USA) and Na_2_HPO_4_ (94046, Sigma–Aldrich) were diluted to equal concentrations of 3 (CPB-S) or 7.5 (CPB-N) mM in Dulbecco’s modified Eagle’s medium (11995065, Thermo Fisher Scientific, Waltham, MA, USA) supplemented with 10% (CPB-S) or 1% (CPB-N) fetal bovine serum (26140079, Thermo Fisher Scientific). For the synthesis of MPBs, stock solutions of MgCl_2_ (E525, VWR, West Chester, PA, USA) and Na_2_HPO_4_ (94046, Sigma–Aldrich) were diluted to equal concentrations of 20 mM in Dulbecco’s modified Eagle’s medium (11995065, Thermo Fisher Scientific) supplemented with 10% fetal bovine serum (26140079, Thermo Fisher Scientific). Following 24 h incubation in cell culture conditions, the medium was centrifuged at 200,000× *g* for 1 h (Optima MAX-XP, 393315, Beckman Coulter, Brea, CA, USA) with the further resuspension of bions in sterile 0.9% NaCl solution. Quantification of the bions was conducted utilizing a microplate spectrophotometer (Multiskan Sky, 51119700DP, Thermo Fisher Scientific) at a 650 nm (OD_650_) wavelength.

### 4.2. Animal Model of Bion-Induced Endothelial Injury

Male Wistar rats weighing 250–300 g and 12–14 weeks of age, provided by the Research Institute for Complex Issues of Cardiovascular Diseases Core Facility, were used for all animal experiments (*n* = 35). Animals were allocated in the polypropylene cages (5 rats per cage) lined with wood chips and had access to the water and food (rat chow) ad libitum. Throughout the duration of the experiment, the standard conditions of the temperature (24 ± 1 °C), relative humidity (55% ± 10%), and a 12 h light/dark cycle were carefully maintained, and the health status of all rats was monitored daily. No randomization was performed to allocate animals to experimental groups or cages. There were no specific inclusion or exclusion criteria. Experiments were performed in a blinded fashion. All procedures were carried out conforming with the European Convention for the Protection of Vertebrate Animals used for Experimental and other Scientific Purposes and were approved by the Ethical Committee of the Research Institute for Complex Issues of Cardiovascular Diseases (ethical approval code 56/2019, approved on 13 May 2019).

To examine whether CPBs were able to induce endothelial injury in the absence of any major cardiovascular risk factors, we conducted regular tail vein injections of CPB-S, CPB-N, or MPB as a control group of the bions (900 µL, OD_650_ = 0.08–0.10) or an equal volume of 0.9% NaCl as a vehicle control (once or thrice a week over 5 weeks, *n* = 5 animals per group) without any surgical intervention.

Upon 5 weeks, all rats were sacrificed by an intraperitoneal injection of a sodium pentobarbital (100 mg/kg body weight) with the excision of the aorta, liver, and spleen and collection of the serum by centrifugation at 2000× *g*. Aortas were then segmented into aortic arch, descending aorta, and aortic bifurcation. The aortic arch and descending aorta were further dissected into two equal (≈1 cm length) segments.

### 4.3. Histological Examination

Halves of the aortic arch and descending aorta, liver, and spleen were fixed in 10% neutral phosphate buffered formalin (06-001, BioVitrum, St. Petersburg, Russian Federation), for 24 h at 4 °C, and washed in tap water over 2 h, dehydrated in ascending ethanol series (70%, 80%, 95%, 1 h per each) and isopropanol (06-002, BioVitrum, 1 h), impregnated in paraffin (06-004, Histomix Extra, BioVitrum, 3 changes, 1 h per each) with the subsequent embedding (06-005, Mister Vax, BioVitrum, 1 h), incubated overnight at 4°C, shortly frozen at −20 °C, and sectioned (Microm HM 325, 902100ER, Thermo Fisher Scientific). To ensure the proper histological examination, we prepared 12 sections (7 µm thickness), evenly distributed across the entire aortic segment, per slide. Sections were then stained with hematoxylin and eosin (05-06004, Bio-Optica, Milan, Italy) for the general examination, alizarin red S (ab142980, Abcam, Cambridge, UK) to detect calcium deposits, and either Russell–Movat’s pentachrome staining kit (010247, Diapath, Martinengo, Italy) to differentiate collagen, smooth muscle tissue, and elastin within aortic segments or van Gieson staining kit (HT25A, Sigma–Aldrich) to assess collagen content in livers and spleens. All stainings were performed according to the corresponding manufacturer’s protocols. Staining results were visualized by light microscopy (AxioImager.A1, Carl Zeiss, Oberkochen, Germany) in a blinded fashion to evaluate the frequency and extent of intimal hyperplasia in aortic segments and liver/spleen fibrosis.

### 4.4. Immunofluorescence Staining

Halves of the aortic arch and descending aorta as well as aortic bifurcations were snap-frozen in optimal cutting temperature compound (Tissue-Tek, 4583, Sakura, Tokyo, Japan) using liquid nitrogen and were then sectioned on a cryostat (Microm HM 525, 387779, Thermo Fisher Scientific) as described above. For the blocking of non-specific protein binding, sections were first incubated in 1% bovine serum albumin (A2153, Sigma–Aldrich) diluted in phosphate buffered saline (PBS, pH 7.4, P4417, Sigma-Aldrich) for 1 h. Samples were then stained with unconjugated (1) mouse anti-CD31 (1:250 dilution, ab24590, Abcam) and either rabbit anti-α-SMA (1:500, ab32575, Abcam) or rabbit anti-CD34 (1:250, ab81289, Abcam); (2) mouse anti-α-SMA (1:500, ab7817, Abcam) and either rabbit anti-collagen IV (1:500, ab6586, Abcam) or rabbit anti-vimentin (1:500, ab92547, Abcam) primary antibodies and incubated at 4 °C for 18 h. Slides were further treated with goat anti-mouse Alexa Fluor 555-conjugated (1:500, ab150114, Abcam) and donkey anti-rabbit Alexa Fluor 488-conjugated secondary antibodies (1:500, ab150073, Abcam) or goat anti-mouse Alexa Fluor 488-conjugated (1:500, ab150113, Abcam) and donkey anti-rabbit Alexa Fluor 555-conjugated secondary antibodies (1:500, ab150074, Abcam) with further incubation for 1 h at room temperature. Washing was performed thrice with PBS. Nuclei were counterstained with 4′,6-diamidino-2-phenylindole (DAPI) for 30 min at RT (10 µg/mL, D9542, Sigma–Aldrich). Coverslips were mounted with ProLong Gold Antifade (P36934, Thermo Fisher Scientific). Slides were examined by confocal laser scanning microscopy in a blinded fashion (LSM 700, Carl Zeiss).

### 4.5. Measurement of Serum Transaminases and Pro-Inflammatory Molecules

Levels of serum aspartate aminotransferase and alanine aminotransferase were determined employing an automated biochemistry analyzer (Konelab 60i, Thermo Fisher Scientific), while concentrations of pro-inflammatory cytokines (IL-1β and MCP-1/CCL2) and acute phase protein (C-reactive protein and ceruloplasmin) were measured using the respective kits of eBioscience (San Diego, CA, USA) and Abcam (BMS630, BMS631INST, ab256398, and ab108820) according to the manufacturer’s protocols.

### 4.6. Gene Expression Profiling and Measurement of Cytokines in Cell Culture Supernatant

Primary human coronary artery endothelial cells (HCAECs, Cell Applications, 300K-05a, San Diego, CA, USA) and human internal thoracic artery endothelial cells (HITAECs, Cell Applications, 308K-05a) were cultured according to the manufacturer’s protocols to the 85–90% confluence (5th passage) in 6 well plates and exposed to 100 µL of MPB, CPB-S, CPB-N (OD_650_ = 0.08–0.10), or PBS (3 wells per group) for 24 h. Total RNA was extracted using PicoPure isolation kit (KIT0204, Thermo Fisher Scientific), quantified (NanoDrop 2000, Thermo Fisher Scientific), and then reverse transcribed (High Capacity cDNA Reverse Transcription Kit, 4368814, Thermo Fisher Scientific). Levels of gene expression were measured by quantitative polymerase chain reaction (qPCR) using the customized primers (500 nM each, Table 1, designed using Primer-BLAST, https://www.ncbi.nlm.nih.gov/tools/primer-blast/) and PowerUp SYBR^®^ Green Master Mix (A25776, Thermo Fisher Scientific) according to the manufacturer’s instructions and a standard cycling mode for primer T_m_ ≥ 60 °C. Technical replicates (*n* = 3 per each sample collected from one well) were performed in all qPCR experiments. Quantification of the mRNA levels was performed by using the 2^−ΔΔ*C*t^ method. Relative transcript levels are expressed as the value relative to the average of three housekeeping genes (*GAPDH*, *ACTB*, and *B2M*) and to PBS group (2^−ΔΔ*C*t^). To measure the levels of IL-1b, IL-6, IL-8, and IL-23, supernatant was collected from the experiment from above (*n* = 11 wells per group), centrifuged at 1000× *g* for 10 min to remove debris and then processed utilizing the respective enzyme-linked immunosorbent assay kits (ab46052, ab178013, ab46032, and ab221837, Abcam) according to the manufacturer’s instructions.

### 4.7. Proteomic Profiling

Primary human coronary artery endothelial cells (HCAECs, Cell Applications, 300K-05a) and human internal thoracic artery endothelial cells (HITAECs, Cell Applications, 308K-05a) were cultured according to the manufacturer’s protocols to the 85–90% confluence (5th passage) in T-75 flasks and exposed to 600 µL of CPB-S (OD_650_ = 0.08–0.10) or PBS for 4 h with the following protein extraction and dot blot profiling for 35 human apoptosis-related proteins employing a respective R&D (Minneapolis, MN, USA) kit (ARY009). Results were visualized using a C-DiGit chemiluminescence blot scanner (LI-COR Biosciences, Lincoln, NE, USA) at a high sensitivity setting (12 min scan).

### 4.8. Statistical Analysis

Statistical analysis was performed using GraphPad Prism 8 (GraphPad Software, San Diego, CA, USA). Regarding descriptive statistics, data are represented by the median, 25th and 75th percentiles, and range. Groups were compared by the Kruskal–Wallis test followed by Dunn’s multiple comparisons test. The *p*-values ≤ 0.05 were regarded as statistically significant.

## Figures and Tables

**Figure 1 ijms-20-05728-f001:**
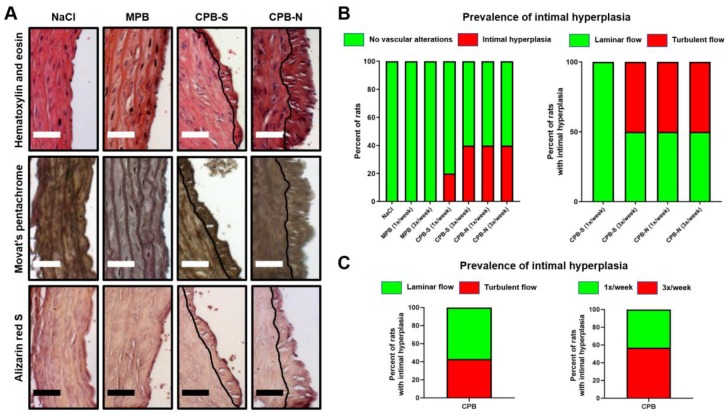
Histological analysis of the intact aortas of normolipidemic Wistar rats who underwent regular (1/3 times a week) tail vein injections of either physiological saline (NaCl), innocuous magnesium phosphate bions (MPB) or spherical (CPB-S)/needle-shaped (CPB-N) calcium phosphate bions. (**A**) Hematoxylin/eosin and Movat’s pentachrome stainings demonstrate visible neointima (demarcated from the media by the black line) in the aortas exposed to CPB-S, or CPB-N but not MPB or NaCl. Negative alizarin red S staining underlines the absence of calcium deposits suggestive of endothelial injury rather than osteogenic transition as a major pathogenic effect of CPBs. Scale bar = 50 µm, close-ups at 200× magnification. (**B**) Quantitative analysis of intimal hyperplasia prevalence reveals that, in contrast to NaCl and MPB, CPB-S and CPB-N cause neointimal formation in 3/10 (30%) and 4/10 (40%) animals, respectively, regardless of the blood flow pattern. (**C**) Pooled quantitative analysis of intimal hyperplasia confirms that the prevalence of intimal hyperplasia does not depend on the blood flow pattern (3/7 neointimal segments in aortic arches/bifurcations characterized by a turbulent flow and 4/7 in the descending aorta having a laminar flow) or regimen of CPB administration (3/7 and 4/7 neointimal segments detected in rats treated with CPBs once and thrice a week, respectively).

**Figure 2 ijms-20-05728-f002:**
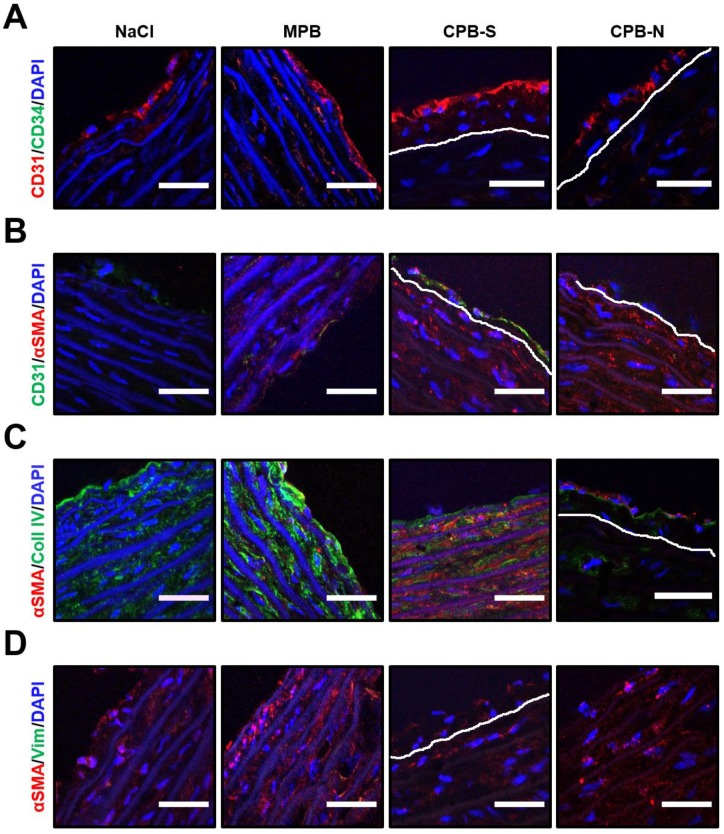
Immunophenotyping of the neointimal segments from initially intact aortas of Wistar rats treated with NaCl, MPB, CPB-S, or CPB-N according to the indicated protocol. (**A**) CD31 (red)/CD34 (green) and (**B**) CD31 (green)/α-smooth muscle actin (αSMA, red) stainings show that both healthy aortic segments and neointima were covered with a monolayer of CD31^+^CD34^-^αSMA^-^ cells attesting to their mature endothelial phenotype. (**C**) αSMA (red)/collagen type IV (coll IV, green) and (**D**) αSMA (red)/vimentin (Vim, green) stainings demonstrate that the αSMA^+^ cells produced extracellular matrix yet not expressing vimentin, a conventional marker of a phenotypic switch. Nuclei were counterstained with 4′,6-diamidino-2-phenylindole (DAPI, blue). White line demarcates the neointima. Scale bar = 50 µm, 630× magnification.

**Figure 3 ijms-20-05728-f003:**
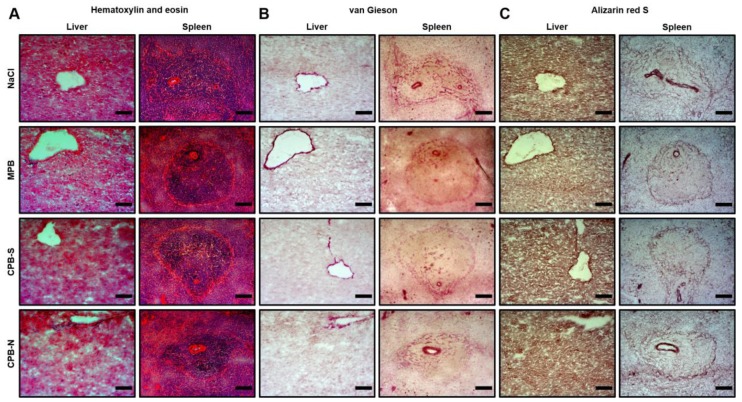
Histological analysis of the liver and spleen of Wistar rats treated with NaCl, MPB, CPB-S, or CPB-N according to the indicated protocol. (**A**) Hematoxylin and eosin, (**B**) van Gieson, and (**C**) alizarin red S stainings show the absence of cellular damage, fibrosis, and mineralization, respectively. Scale bar = 100 µm, 200× magnification.

**Figure 4 ijms-20-05728-f004:**
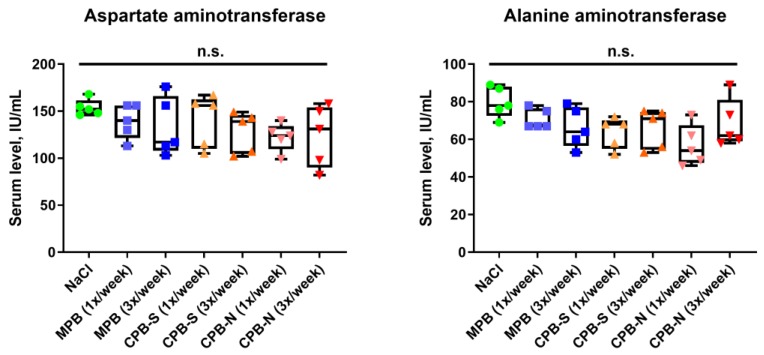
Quantification of serum transaminases in Wistar rats treated with NaCl, MPB, CPB-S, or CPB-N according to the indicated protocol. Box-and-whisker plots combined with a univariate scatterplot, where each dot represents a serum sample collected from one rat. Whiskers indicate range, box bounds indicate the 25th and 75th percentiles, and center lines indicate the median. Dunn’s multiple comparisons test, n.s. is for non-significant.

**Figure 5 ijms-20-05728-f005:**
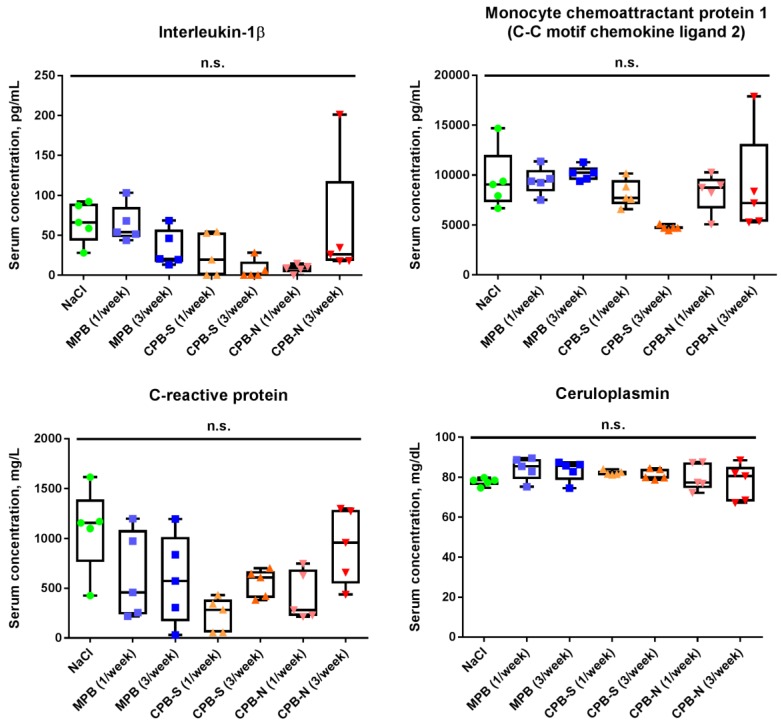
Quantification of serum pro-inflammatory cytokines (interleukin-1β and monocyte chemoattractant protein 1/C-C motif chemokine ligand 2) and acute phase proteins (C-reactive protein and ceruloplasmin) in Wistar rats treated with NaCl, MPB, CPB-S, or CPB-N according to the indicated protocol. Box-and-whisker plots combined with a univariate scatterplot, where each dot represents a serum sample collected from one rat. Whiskers indicate range, box bounds indicate the 25th and 75th percentiles, and center lines indicate the median. Dunn’s multiple comparisons test, n.s. is for non-significant.

**Figure 6 ijms-20-05728-f006:**
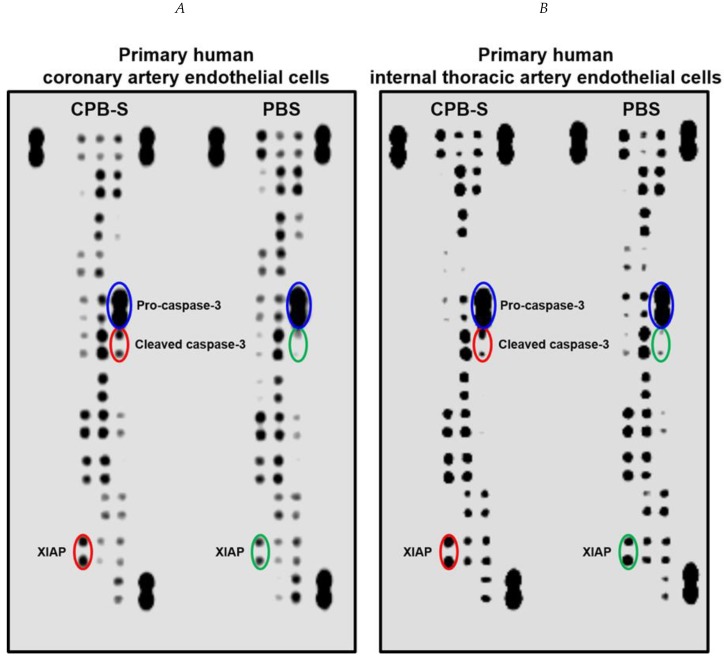
Proteomic profiling of apoptosis pathways in (**A**) HCAECs and (**B**) HITAECs upon exposure to CPB-S. Note the increase in cleaved caspase-3 in relation to procaspase-3 and the elevation of XIAP in both types of endothelial cells treated with CPB-S.

**Figure 7 ijms-20-05728-f007:**
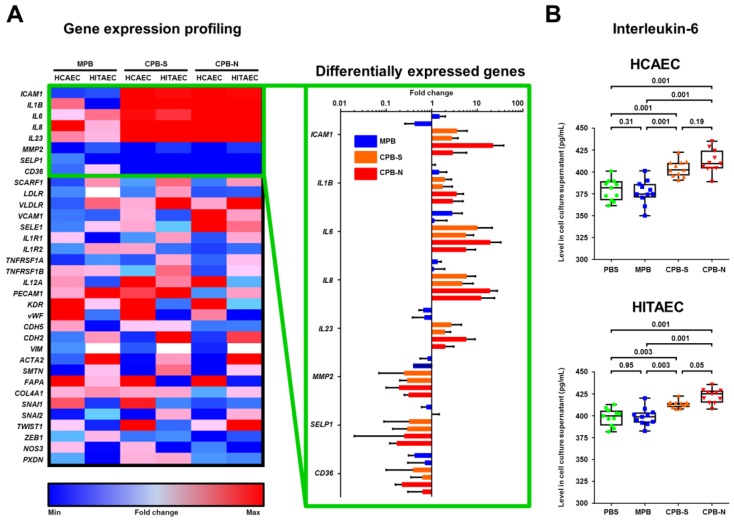
Exposure of HCAECs and HITAECs to CPBs induces the release of pro-inflammatory cytokines by endothelial cells. (**A**) HCAECs and HITAECs were cultured in the presence of MPB, CPB-S, or CPB-N for 24 h, and the total RNA was extracted following expression profiling for the indicated genes (*n* = 3 wells per group). Heat map shows the differentially expressed genes among groups. Statistically significant differentially expressed genes (fold change >2) are presented in the box to the right. (**B**) HCAECs and HITAECs were cultured in the presence of MPB, CPB-S, or CPB-N for 24 h. Conditioned media were collected and profiled for interleukin-6 using ELISA (*n* = 11 wells per group). Each dot represents one well of culture plate. Whiskers indicate range, box bounds indicate the 25th–75th percentiles, and center lines indicate the median. The *p*-values are provided above the boxes. Dunn’s multiple comparisons test was used.

**Table 1 ijms-20-05728-t001:** Primers for qPCR and parameters of the reaction.

Gene	Primers	*R* ^2^	Efficiency, %
*ACTB*	F: 5′-CATCGAGCACGGCATCGTCA-3′ R: 5′-TAGCACAGCCTGGACAGCAAC-3′	0.996	92.834
*GAPDH*	F: 5′-AGCCACATCGCTCAGACAC-3′ R: 5′-GCCCAATACGACCAAATCC-3′	0.994	105.854
*B2M*	F: 5′-TCCATCCGACATTGAAGTTG-3′ R: 5′-CGGCAGGCATACTCATCTT-3′	0.990	76.601
*SCARF1*	F: 5′-CCGATCAGACCTCAAGGACAG-3′ R: 5′-CCCAGGGTAGCTTGTGGGA-3′	0.997	97.848
*CD36*	F: 5′-GGCTGTGACCGGAACTGTG-3′ R: 5′-AGGTCTCCAACTGGCATTAGAA-3′	0.992	89.779
*LDLR*	F: 5′-ACGGCGTCTCTTCCTATGACA-3′ R: 5′-CCCTTGGTATCCGCAACAGA-3′	0.991	97.402
*VLDLR*	F: 5′-AGAAAAGCCAAATGTGAACCCT-3′ R: 5′-CACTGCCGTCAACACAGTCT-3′	0.990	90.926
*VCAM1*	F: 5′-CGTCTTGGTCAGCCCTTCCT-3′ R: 5′-ACATTCATATACTCCCGCATCCTTC-3′	0.987	86.482
*ICAM1*	F: 5′-TTGGGCATAGAGACCCCGTT-3′ R: 5′-GCACATTGCTCAGTTCATACACC-3′	0.993	107.375
*PECAM1*	F: 5′-TGGCGCATGCCTGTAGTA-3′ R: 5′-TCCGTTTCCTGGGTTCAA-3′	0.987	85.560
*SELE1*	F: 5′-GCACAGCCTTGTCCAACC-3′ R: 5′-ACCTCACCAAACCCTTCG-3′	0.990	95.994
*SELP1*	F: 5′-ATGGGTGGGAACCAAAAAGG-3′ R: 5′-GGCTGACGGACTCTTGATGTAT-3′	0.989	94.008
*CDH5*	F: 5′-AAGCGTGAGTCGCAAGAATG-3′ R: 5′-TCTCCAGGTTTTCGCCAGTG-3′	0.996	91.622
*IL1R1*	F: 5′-GGCTGAAAAGCATAGAGGGAAC-3′ R: 5′-CTGGGCTCACAATCACAGG-3′	0.988	98.004
*IL1R2*	F: 5′-TGGCACCTACGTCTGCACTACT-3′ R: 5′-TTGCGGGTATGAGATGAACG-3′	0.988	89.115
*TNFRSF1A*	F: 5′-CCAGGAGAAACAGAACACCGT-3′ R: 5′-AAACCAATGAAGAGGAGGGATAA-3′	0.995	87.900
*TNFRSF1B*	F: 5′-GTCCACACGATCCCAACAC-3′ R: 5′-CACACCCACAATCAGTCCAA-3′	0.981	96.901
*NOS3*	F: 5′-GTGATGGCGAAGCGAGTGAAG-3′ R: 5′-CCGAGCCCGAACACACAGAAC-3′	0.992	90.406
*PXDN*	F: 5′-AGCCAGCCATCACCTGGAAC-3′ R: 5′-TTCCGGGCCACACACTCATA-3′	0.987	91.478
*IL1B*	F: 5′-TGGCTTATTACAGTGGCAATG-3′ R: 5′-GTGGTGGTCGGAGATTCG-3′	0.997	107.422
*IL6*	F: 5′-GGCACTGGCAGAAAACAACC-3′ R: 5′-GCAAGTCTCCTCATTGAATCC-3′	0.992	97.406
*IL8*	F: 5′-CAGAGACAGCAGAGCACAC-3′ R: 5′-AGTTCTTTAGCACTCCTTGGC-3′	0.992	105.816
*IL12A*	F: 5′-GCCTTCACCACTCCCAAAAC-3′ R: 5′-TGTCTGGCCTTCTGGAGCAT-3′	0.989	89.617
*IL23*	F: 5′-CTCAGGGACAACAGTCAGTTC-3′ R: 5′-ACAGGGCTATCAGGGAGCA-3′	0.981	92.297
*VWF*	F: 5′-CCTTGACCTCGGACCCTTATG-3′ R: 5′-GATGCCCGTTCACACCACT-3′	0.996	107.880
*KDR*	F: 5′-TGCCTACCTCACCTGTTTC-3′ R: 5′-GGCTCTTTCGCTTACTGTTC-3′	0.980	90.308
*FAP*	F: 5′-TCAACTGTGATGGCAAGAGCA-3′ R: 5′-TAGGAAGTGGGTCATGTGGGT-3′	0.980	107.368
*ACTA2*	F: 5′-GTGTTGCCCCTGAAGAGCAT-3′ R: 5′-GCTGGGACATTGAAAGTCTCA-3′	0.982	107.327
*SMTN*	F: 5′-GGGATCGTGTCCACAAGTTCA-3′ R: 5′-GCTACTCCTCGTTGCTCCTT-3′	0.999	96.724
*CDH2*	F: 5′-GCTTCTGGTGAAATCGCATTA-3′ R: 5′-AGTCTCTCTTCTGCCTTTGTAG-3′	0.994	93.226
*VIM*	F: 5′-CGCCAGATGCGTGAAATGG-3′ R: 5′-ACCAGAGGGAGTGAATCCAGA-3′	0.989	90.249
*COL4A1*	F: 5′-GGACTACCTGGAACAAAAGGG-3′ R: 5′-GCCAAGTATCTCACCTGGATCA-3′	0.992	91.030
*MMP2*	F: 5′-CCGTGTTTGCCATCTGTTTTAG-3′ R: 5′-AGGTTCTCTTGCTGTTTACTTTGGA-3′	0.981	94.465
*SNAI1*	F: 5′-CAGACCCACTCAGATGTCAAGAA-3′ R: 5′-GGGCAGGTATGGAGAGGAAGA-3′	0.993	94.588
*SNAI2*	F: 5′-ACTCCGAAGCCAAATGACAA-3′ R: 5′-CTCTCTCTGTGGGTGTGTGT-3′	0.986	87.981
*TWIST1*	F: 5′-GTCCGCAGTCTTACGAGGAG-3′ R: 5′-GCTTGAGGGTCTGAATCTTGCT-3′	0.996	96.188
*ZEB1*	F: 5′-GATGATGAATGCGAGTCAGATGC-3′ R: 5′-ACAGCAGTGTCTTGTTGTTGT-3′	0.983	90.857

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
