# Peer review of "Calcium Phosphate Bions Cause Intimal Hyperplasia in Intact Aortas of Normolipidemic Rats through Endothelial Injury"

_ijms, 2019, doi:10.3390/ijms20225728_

Round 1

Reviewer 1 Report

The authors addressed the capability of CPB to induce intimal hyperplasia in normolipidemic conditions and absence of additional cardiovascular risk factors, and if it was affected by CPB shape and blood flow pattern in aortic segments. It was suggested that CPB were able to cause typical intimal hyperplasia per se through an injury of initially intact endothelium.

Although the experimental part was well prepared and adequately performed, the discussion section needs major revision. Namely, although it is always necessary to briefly describe the obtained results, a much better translation of presented findings to possible clinical consequences is more than necessary in this case. Hence, the authors are strongly suggested to revise the discussion section in a more comprehensive manner.

Author Response

We agree with that the clinical relevance of our results and the entire CPP topic needed to be much better described. Hence, we attempted to review the available literature thoroughly and added discussion and perspectives of clinical translation.

We sincerely thank the reviewer for his/her valuable suggestions.

Reviewer 2 Report

Shishkova and coworkers provide a rat study on the endothelial injury caused by CPB, resulting in intimal hyperplasia. Overall, the manuscript is nicely written and the topic fits with the International Journal of Molecular Science. They performed regular injections of CPB into the tail vein of normolipidemic Witstar rats to study intimal hyperlplasia, liver and spleen injury. The message of this paper ist hat CPB induces the formation of neointima in rat aortas, which was confirmed by immunofluorescence and histology. This was independent of liver damange and splenic injruy, as confirmed by histologies and unchanged liver enzyme levels. Also the serum levels of cytokines and acute phase proteins were unchanged, so the question remains what is the cause of increased intimal hyperplasia following CPB treatment. In the reviewer’s view, in spite of having a nice set of descriptive data, the autors do not resolve cause and consequence of this vascular phenotype. In ordert to move from a merely descriptive phenotyping study to a mechanism, I suggest that the autors should pinpoint the exact mechanism of neointimal hyperplasia induced by CPB treatment and then submit a revised manuscript version that resolves this issue.

Author Response

We agree that the identification of the exact mechanism for the promotion of intimal hyperplasia development is mandatory to properly assess the causes of the vascular phenotype triggered by CPB. Hence, we performed in vitro functional experiments on primary human coronary artery and internal thoracic artery endothelial cells, evaluating their gene expression profile, expression of multiple apoptosis-related proteins, and release of a major pro-inflammatory cytokine IL-6 upon treatment with CPB. 

We found that CPB induce regulated death of primary human endothelial cells involving intrinsic apoptosis cascade (i.e., cleavage of caspase-9 and caspase-3 and reciprocal upregulation of XIAP) and promote release of IL-6 through the upregulation of the respective gene in addition to some other cytokine genes (IL1B, IL8, and IL23). However, levels of IL-1b and IL-23 in cell culture supernatant were negligible while the concentrations of IL-8 did not differ significantly between the groups.

These results, taken together with the findings of other groups from vascular smooth muscle cells [Refs. 35-37] and macrophages [Refs. 34, 38, 39], suggest that CPB evoke local rather than systemic inflammation, thereby altering the paracrine signaling between different populations of vascular cells and contributing to the development of pathological microenvironment. The mechanism behind the neointima formation seems to be the combination of crude endothelial injury and stimulation of pro-inflammatory signaling in situ.

We sincerely thank the reviewer for his/her valuable suggestions.

Reviewer 3 Report

The submitted manuscript titled "Calcium phosphate bions cause intimal hyperplasia in intact aortas of normolipidemic rats through endothelial injury" by Daria Shishkova et al. is a research article that aims to investigate the effects of calcium phosphate bions on intact rat arteries. The findings reported and further developed in its discussion provide a certain insight into the arterial calcification and are of scientific interest for this specific research field, while also falling within the scope of the journal. This is a clear and interesting article analyzing animal model. Strikingly, the manuscript also highlights mechanisms of endothelial dysfunction.  The paper is well-documented, and contains an interesting discussion. 

Author Response

We sincerely thank the reviewer for the consideration of our paper to be published.

Reviewer 4 Report

Have authors measured the serum concentration of CBP in the experimental animals? Figure 1A: The quality of the histological images is poor, and several sectioning artifacts are present. The signs of neointima formation are not clear in the presented image. It appears the images were not taken under the same magnification. Please replace with high quality image and label the neointima. Figure 1B: The authors pooled the data from two different injection frequencies in this chart, which may lead to misinterpretation. Please separate the results similar to that in Figure 5. Please also separate the data obtained from laminar flow and turbulent flow. Figure 2: Please label the neointima in the images. In panel C, collagen IV was expressed in NaCl, MBP and CBP-S, but was low in CBP-N. What was the authors interpretation on this observation? aSMA was not expressed in controls in panel B and C, but was expressed in controls in panel D. What caused this inconsistency? Figure 4: Please group the data similar to that in Figure 5.

Author Response

We agree with all the reviewer's comments and considerably revised the paper. 

The serum concentration of CPB in experimental animals has not been measured due to their rapid clearance from the circulation (around 25 minutes).

We reworked the Figure 1A by means of removing sectioning artifacts, replacing the images, and labeling the neointima. Yet, the images have been indeed obtained under the same magnification. We also reworked Figure 1B and Figure 4 according to the reviewer's suggestion to separate the data across all groups and blood flow patterns as it was done in Figure 5. 

The neointima is now labeled in Figure 2. Inconsistencies between collagen IV and alpha smooth muscle actin stainings may be caused by poor tissue preservation in these samples due to a long-term storage. Unfortunately, other samples have been utilised for other purposes to the time of the immunofluorescence stainings. Our point is that these are the only slides affected by this condition and these artifacts are not related to the neointimal phenotyping.

We sincerely thank the reviewer for his/her valuable suggestions.

Round 2

Reviewer 1 Report

/

Author Response

The introduction now provides sufficient background and includes all relevant references. We think we have addressed all the issues raised by the reviewer.

Reviewer 4 Report

According to the Materials and Methods, 5 rats were used in each group. However, in the revised Figure 1B, 10% of 5 rats in group CPB-S (1x/week) were found to have intimal hyperplasia. How did the authors get this ratio? 

Author Response

We have corrected Figure 1 as it was a calculation mistake occurred during the entering the results into the GraphPad. As the numbers of rats with intimal hyperplasia are 1 or 2 out of 5 depending on the experimental group (revised Figure 1B), the exact proportion is 20 or 40%, of course. We thank the reviewer for this notice.